# Caspase-1 cleaves Bid to release mitochondrial SMAC and drive secondary necrosis in the absence of GSDMD

Rosalie Heilig, Marisa Dilucca, Dave Boucher ⓘ, Kaiwen W Chen ⓘ, Dora Hancz, Benjamin Demarco ⓘ, Kateryna Shkarina, Petr Broz ⓘ

Caspase-1 drives a lytic inflammatory cell death named pyroptosis by cleaving the pore-forming cell death executor gasdermin-D (GSDMD). *Gsdmd* deficiency, however, only delays cell lysis, indicating that caspase-1 controls alternative cell death pathways. Here, we show that in the absence of GSDMD, caspase-1 activates apoptotic initiator and executioner caspases and triggers a rapid progression into secondary necrosis. GSDMD-independent cell death required direct caspase-1–driven truncation of Bid and generation of caspase-3 p19/p12 by either caspase-8 or caspase-9. tBid-induced mitochondrial outer membrane permeabilization was also required to drive SMAC release and relieve inhibitor of apoptosis protein inhibition of caspase-3, thereby allowing caspase-3 auto-processing to the fully active p17/p12 form. Our data reveal that cell lysis in inflammasome-activated *Gsdmd*-deficient cells is caused by a synergistic effect of rapid caspase-1–driven activation of initiator caspases-8/-9 and Bid cleavage, resulting in an unusually fast activation of caspase-3 and immediate transition into secondary necrosis. This pathway might be advantageous for the host in counteracting pathogen-induced inhibition of GSDMD but also has implications for the use of GSDMD inhibitors in immune therapies for caspase-1–dependent inflammatory disease.

## Introduction

Inflammasomes are cytosolic signalling platforms assembled after the recognition of host- or pathogen-derived danger signals by cytosolic pattern recognition receptors, such as pyrin, AIM2, and members of the Nod like receptor (NLR) protein family (Broz & Dixit, 2016). These complexes serve as activation platforms for caspase-1, the prototypical inflammatory caspase. Active caspase-1 cleaves the pro-inflammatory cytokines IL-1β and IL-18 to their mature bioactive form and induces a lytic form of cell death known as pyroptosis, by processing the cell death executor gasdermin-D (GSDMD) (Kayagaki et al, 2015; Shi et al, 2015). Caspase cleavage

at the residue D276 in mouse (D275 in human) removes the inhibitory GSDMD^CT and allows GSDMD^NT to translocate to cellular membranes and form permeability pores, which disrupt ion homeostasis and the electrochemical gradient (Kayagaki et al, 2015; Shi et al, 2015; Aglietti et al, 2016; Ding et al, 2016; Liu et al, 2016; Sborgi et al, 2016). GSDMD is also cleaved by caspase-11 in mice and by caspase-4 and caspase-5 in humans, which are activated by the so-called noncanonical inflammasome pathway in response to LPS stemming from infections with cytosolic Gram-negative bacteria (Kayagaki et al, 2011, 2013; Hagar et al, 2013; Shi et al, 2014). Uncontrolled inflammasome activation by gain-of-function mutations in inflammasome receptors or in the context of sterile inflammatory disease has been linked to a number of hereditary and acquired inflammatory diseases, such as cryopyrin-associated periodic syndrome (Muckle–Wells syndrome), but also gout, Alzheimer's disease, and atherosclerosis (Masters et al, 2009). It is, thus, of high interest to target and inhibit inflammasome assembly or downstream effector processes such as GSDMD pore formation and IL-1β release.

Although *Gsdmd* deficiency results in complete abrogation of caspase-11 (-4)–induced lytic cell death, it only delays caspase-1–induced cell lysis (He et al, 2015; Kayagaki et al, 2015). Caspase-1 activation in *Gsdmd*⁻/⁻ cells correlates with high levels of caspase-3/7 and caspase-8 activity, but whether these apoptotic caspases trigger lysis of *Gsdmd*-deficient cells after caspase-1 activation has not been proven (He et al, 2015), and activation of apoptotic caspases has been observed to occur even in inflammasome-activated WT cells (Lamkanfi et al, 2008; Sagulenko et al, 2018). The lytic death of *Gsdmd*⁻/⁻ cells is also in contrast to the notion that apoptosis is non-lytic and, thus, immunologically silent. However, it is also known that prolonged apoptotic caspase activity will result in apoptotic cells losing membrane integrity, a process termed "secondary necrosis." Apoptosis is executed by caspase-3/-7, which themselves are activated by either caspase-8 (extrinsic apoptosis pathway) or caspase-9 (intrinsic or mitochondrial apoptosis pathway). Ligation of death receptors at the plasma membrane (FasR, tumor necrosis factor receptor, and Trail) results in the assembly of the death-inducing signalling complex or tumor necrosis factor receptor complex IIa/b, which activates caspase-8, the initiator caspase of the extrinsic pathway. In type-I cells, caspase-8

Department of Biochemistry, University of Lausanne, Epalinges, Switzerland

Correspondence: petr.broz@unil.ch

activity is sufficient to activate the executioner caspases, whereas in type-II cells, caspase-8 requires activation of the intrinsic pathway in addition (Jost et al, 2009). Here, caspase-8 cleaves the Bcl-2 family protein Bid to generate a truncated version (tBid), which triggers Bax/Bak–induced mitochondrial outer membrane permeabilization (MOMP). MOMP results in the release of second mitochondria-derived activator of caspases (SMAC), ATP, and cytochrome c to promote intrinsic apoptosis via formation of the apoptosome. This complex consists of apoptotic protease-activating factor 1 (APAF1), cytochrome c, ATP, and caspase-9 and serves as an activation platform for caspase-9, which in turn cleaves caspase-3. Apoptosis is a tightly regulated process, and disturbance of the equilibrium of cytosolic pool of pro- and anti-apoptotic Bcl-2 family proteins can result in MOMP, apoptosis induction, and cell death (Riley, 2018; Vince et al, 2018). To prevent accidental activation of apoptosis, inhibitor of apoptosis proteins (IAPs), in particular X-linked inhibitor of apoptosis protein (XIAP), suppresses caspase-3/7 and caspase-9 activation by direct binding to the caspases via baculovirus IAP repeat (BIR) domains (Roy et al, 1997; Takahashi et al, 1998; Bratton et al, 2002; Scott et al, 2005). SMAC, which is released during MOMP, antagonizes IAPs, thus removing the brake on caspase auto-processing and allowing full activity of the executioner caspases and apoptotic cell death (Du et al, 2000; Verhagen et al, 2000; Wilkinson et al, 2004).

Here, we investigate the mechanism that induces lytic cell death after caspase-1 activation in *Gsdmd*-deficient cells. We show that cell death in *Gsdmd*$^{-/-}$ macrophages requires caspase-1, Bid-dependent mitochondrial permeabilization, and the executioner caspase-3. Remarkably, *Gsdmd*-deficient cells form apoptotic blebs and bodies only transiently, before shifting rapidly to a necrotic phenotype that is characterized by extensive membrane ballooning. Unexpectedly, we found that Bid cleavage and subsequent MOMP is driven directly by caspase-1 independently of caspase-8, although high levels of cleaved caspase-8 p18 are found in inflammasome-activated *Gsdmd*-deficient cells. Upon investigating the steps downstream of MOMP, we observed that knocking-out *Casp9* in *Gsdmd*$^{-/-}$ cells had only a small effect on cell death, whereas removing both *Casp8* and *Casp9* abrogated GSDMD-independent cell death. The redundancy in caspase-8 and caspase-9 requirement was explained by the observation that either caspase was sufficient to process caspase-3 between the large and small catalytic domains, thereby generating the intermediate caspase-3 p19 and p12 fragments. Caspase-1–dependent Bid cleavage and SMAC release are then required to remove IAP inhibition, thereby allowing auto-cleavage of caspase-3 to the p17/p12 fragments and full caspase activation (Kavanagh et al, 2014). Thus, cell lysis in the absence of GSDMD is driven by the synergistic effect of both rapid caspase-1–driven activation of initiator caspases-8/-9 and Bid cleavage, which results in an unusually fast activation of caspase-3 and immediate transition into secondary necrosis.

# Results

### Canonical inflammasomes trigger a rapid secondary necrosis in the absence of GSDMD

The canonical and noncanonical inflammasome pathways converge on the caspase-dependent cleavage and activation of the pyroptosis executor GSDMD (Kayagaki et al, 2015; Shi et al, 2015). However, although GSDMD is essential for lytic cell death (pyroptosis) after LPS-induced noncanonical inflammasome activation (Fig S1A), *Gsdmd* deficiency only delays cell lysis after engagement of canonical inflammasome receptors, such as AIM2 (Figs 1A and S1B–D), NLRC4, and NLRP3 (Figs 1A and S1B–D) (Kayagaki et al, 2015). The absence of caspase-1 and caspase-11 in primary BMDMs, by contrast, showed a much stronger reduction in lactate dehydrogenase (LDH) release and propidium iodide (PI) influx, and *Asc* deficiency completely abrogated cell lysis after AIM2 or NLRP3 activation, in line with the reported Apoptosis-associated speck-like protein containing a CARD (ASC)-dependent activation of apoptosis in absence of caspase-1 (Pierini et al, 2012; Man et al, 2013; Sagulenko et al, 2013; Chen et al, 2015; Vajjhala et al, 2015).

We next tested a number of cell death inhibitors for their ability to block cell lysis in *Gsdmd*$^{-/-}$ immortalized BMDMs (iBMDMs) transfected with poly(dA:dT), an activator of the AIM2 inflammasome (Fig S2). Neither 7-Cl-O-Nec1 (RIPK1 kinase inhibitor) nor GSK872 (RIPK3 kinase inhibitor) were able to delay cell death in *Gsdmd*$^{-/-}$ iBMDMs, thereby excluding a role for necroptosis or complex IIb-dependent apoptosis, which require the kinase activity of RIPK3 or RIPK1, respectively (Cho et al, 2009; He et al, 2009; Zhang et al, 2009; Feoktistova et al, 2011; Tenev et al, 2011). Similarly, we ruled out the involvement of calpains, calcium-dependent proteases (PD 150606 and Calpeptin), or cathepsins (pan-cathepsin inhibitor K777), which were previously shown to induce apoptosis through a caspase-3–dependent or caspase-3–independent mechanisms (Stennicke et al, 1998; Chwieralski et al, 2006; Momeni, 2011). Finally, we also tested if caspase inhibitors delayed death in *Gsdmd*$^{-/-}$ or WT iBMDMs. Remarkably, we found that whereas the pan-caspase inhibitor VX765 delayed PI uptake in both WT and *Gsdmd*$^{-/-}$ poly(dA:dT) transfected cells, the specific caspase-3/-7 inhibitor I only blocked cell death in *Gsdmd*$^{-/-}$ but not in WT cells (Figs 1B and S2). VX765 failed to prevent cell death in WT cells at later time points in accordance with previous studies that showed pyroptosis is difficult to block pharmacologically (Schneider et al, 2017).

Although this suggested that apoptotic executioner caspases were necessary for cell death in *Gsdmd*-deficient cells but dispensable for cell death in WT cells, the speed by which *Gsdmd*-deficient cells underwent apoptosis and subsequently cell lysis was remarkable. *Gsdmd*$^{-/-}$ BMDMs displayed DNA laddering and processing of caspase-3 to the mature p17 fragment within 1 h after poly(dA:dT) transfection, which was faster than even the highest concentrations of either extrinsic or intrinsic apoptosis stimuli tested (Fig 1C and D). It is noteworthy that the highest concentration regularly used to induce apoptosis is yet 20 times lower than the concentration used in our study (Vince et al, 2018). Phenotypically, this rapid activation of caspase-3 resulted in a very fast lytic cell death as measured by PI influx (Fig 1E) and morphological analysis (Fig 1F). Of note, inflammasome-stimulated *Gsdmd*$^{-/-}$ BMDMs initiated membrane blebbing and apoptotic body formation initially, but rapidly lost this morphology and transitioned into a necrotic state, characterized by extensive membrane ballooning (Fig 1F), similarly to the end-stage of GSDMD-induced pyroptosis (Fig S3A–C). We conclude that inflammasome activation in the absence of GSDMD results in rapid cell lysis, which we propose to refer to as "GSDMD-independent secondary necrosis" to reflect both the

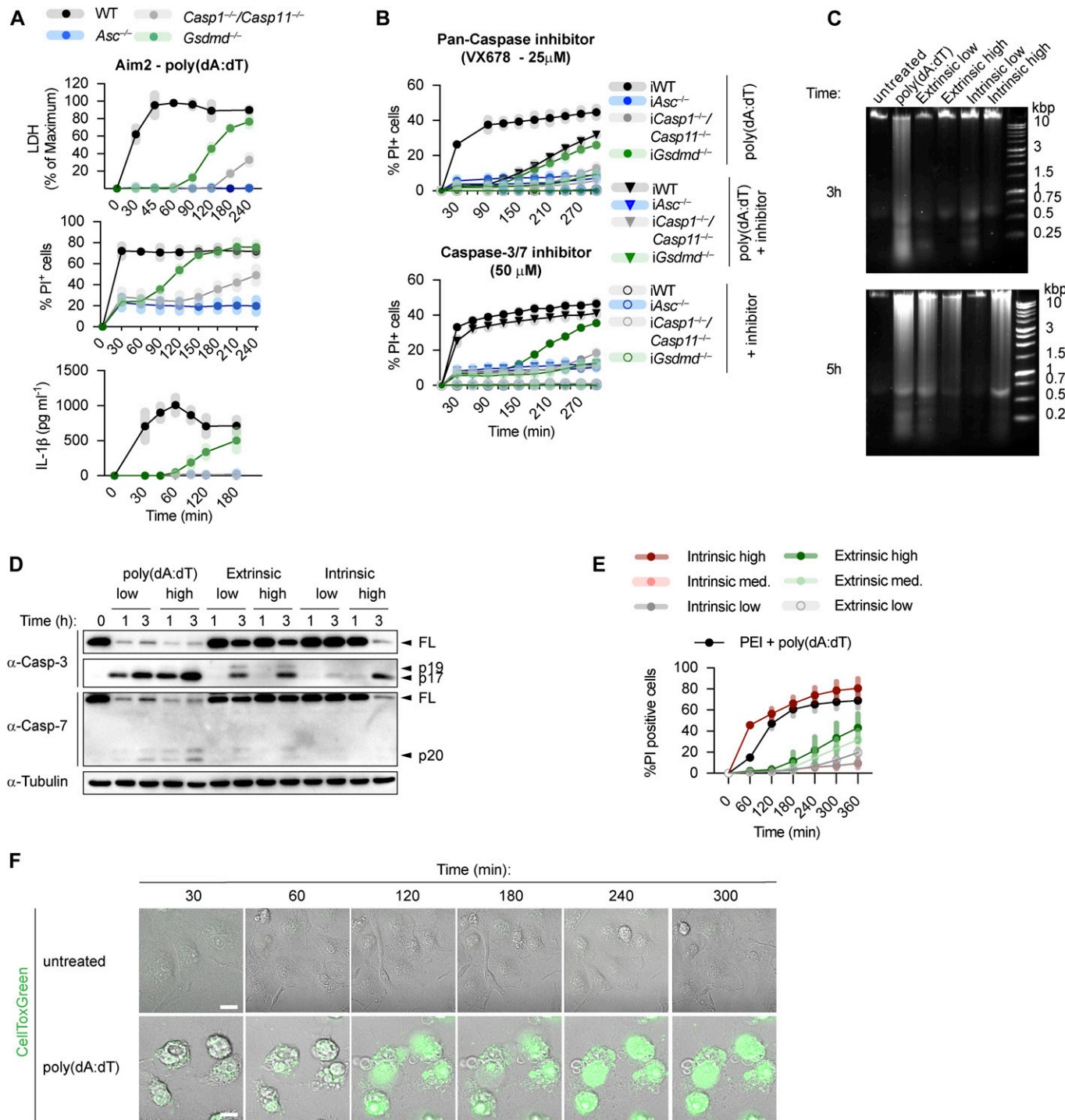

**Figure 1. Canonical inflammasome activation *Gsdmd*-deficient macrophages results in rapid secondary necrosis.**
**(A, B)** LDH release, PI influx, and IL-1β release from LPS-primed WT, *Asc*⁻/⁻, *Casp1*⁻/⁻/*Casp11*⁻/⁻, and *Gsdmd*⁻/⁻ primary or immortalized BMDMs (BMDMs and iBMDMs) after transfection of poly(dA:dT) in the absence or presence of the indicated inhibitors. **(C, D, E)** DNA cleavage, PI influx, and immunoblots showing caspase-3/-7 processing from LPS-primed *Gsdmd*⁻/⁻ BMDMs transfected with poly(dA:dT) or treated with 100 ng/ml TNF-α plus 10, 5, or 1 μM AZD5582 (extrinsic apoptosis) or 1 μM ABT-737 plus 10, 1, or 0.5 μM S63845 (intrinsic apoptosis). **(F)** Confocal images of LPS-primed *Gsdmd*⁻/⁻ BMDMs transfected with poly(dA:dT) or left untreated and stained with CellTox Green (green). Scale bar = 10 μM. Graphs show mean ± SD. Data and blot are representative of at least three independent experiments.

rapid transition to the necrotic state and the requirement for the activity of the apoptotic executioner caspases-3/-7.

## GSDMD-independent secondary necrosis is mainly driven by caspase-3

We next investigated which executioner caspase was required for GSDMD-independent secondary necrosis after caspase-1 activation. High levels of caspase-3/-7 activity was detected in poly(dA:dT)–transfected and *Salmonella*-infected *Gsdmd*$^{-/-}$ and to a lesser degree in *Casp1*$^{-/-}$/*Casp11*$^{-/-}$ BMDMs, whereas WT or *Asc*$^{-/-}$ BMDMs showed minimal to no activity (Figs 2A and S4A). Because both caspase-3 and caspase-7 cleave the DEVD peptidic substrate, we next determined which executioner caspase was cleaved in *Gsdmd*$^{-/-}$ cells but found that both caspase-3 and caspase-7 were rapidly cleaved (Fig 2B). Although cleaved caspase-7 was detected in both WT and *Gsdmd*-deficient cells, only *Gsdmd*$^{-/-}$ cells display detectable caspase-3/-7 activity and caspase-3 cleavage (Fig 2A). We, therefore, hypothesized that caspase-3 must account for the DEVDase activity in *Gsdmd*$^{-/-}$ BMDMs.

To confirm our hypothesis genetically, we used CRISPR/Cas9 genome engineering to delete either *Casp3* or *Casp7*, or both *Casp-3/7* in *Gsdmd*$^{-/-}$ BMDMs (Fig S4B) and determined the impact of the deletion on GSDMD-independent secondary necrosis after AIM2 inflammasome activation (Figs 2C and S4C). *Gsdmd*$^{-/-}$/*Casp3*$^{-/-}$ as well as *Gsdmd*$^{-/-}$/*Casp3*$^{-/-}$/*Casp7*$^{-/-}$ iBMDMs were strongly protected against cell death after poly(dA:dT) transfection, whereas *Casp7* single deficiency did not provide protection, despite previous reports that caspase-3 and caspase-7 function in a redundant manner (Figs 2C and S4C) (Walsh et al, 2008; Lamkanfi & Kanneganti, 2010). Caspase-7 appeared to mainly contribute to the cell death observed in *Gsdmd*$^{-/-}$/*Casp3*$^{-/-}$ iBMDMs, as these had higher LDH levels than *Gsdmd*$^{-/-}$/*Casp3*$^{-/-}$/*Casp7*$^{-/-}$ iBMDMs (Fig 2C). These data were further corroborated by knockdown of caspase-3 or caspase-7 in *Gsdmd*$^{-/-}$ iBMDMs (Fig S4D). Finally, we also examined cell morphology after poly(dA:dT) transfection. *Casp7* knockout in *Gsdmd*$^{-/-}$ iBMDMs failed to reduce necrotic features and cell lysis, whereas *Gsdmd*$^{-/-}$/*Casp3*$^{-/-}$ and *Gsdmd*$^{-/-}$/*Casp3*$^{-/-}$/*Casp7*$^{-/-}$ iBMDMs remained alive and intact (Fig 2D) at 3 h posttreatment. In summary, these results demonstrate that although both executioner caspases are cleaved during cell death, it is caspase-3 that drives GSDMD-independent secondary necrosis in inflammasome-activated cells.

Because caspase-3 was shown to cleave gasdermin-E (GSDME), another member of the gasdermin family, and GSDME was proposed to drive secondary necrosis during prolonged apoptosis, we asked whether lack of GSDMD drives an alternative cell death pathway via caspase-3–mediated GSDME cleavage and pore formation. We, thus, measured LDH release and PI influx in WT, *Gsdmd*$^{-/-}$, *Gsdme*$^{-/-}$, and *Gsdmd*$^{-/-}$/*Gsdme*$^{-/-}$ BMDMs upon activation of the AIM2 inflammasome (Figs 2E and S5A). Surprisingly, although GSDME was cleaved in *Gsdmd*$^{-/-}$ at 1 h post-poly(dA:dT) transfection, we did not find a contribution of GSDME to cell death in *Gsdmd*$^{-/-}$ BMDMs because double *Gsdmd*/*Gsdme*-deficiency did not confer any additional protection (Figs 2E and S5A and B). Furthermore, BMDMs lacking only GSDME were comparable with WT

BMDMs, overall suggesting that GSDME does neither contribute to pyroptosis nor GSDMD-independent necrosis.

## Caspase-1 is required to cause GSDMD-independent secondary necrosis in inflammasome-activated cells

Because the ASC speck has been reported to control activation of apoptotic caspases independently of caspase-1 (Lee et al, 2018; Mascarenhas et al, 2017; Pierini et al, 2012; Sagulenko et al, 2013; Schneider et al, 2017; Van Opdenbosch et al, 2017), we next generated *Gsdmd*$^{-/-}$/*Casp1*$^{-/-}$ BMDMs to determine if caspase-1 was required for GSDMD-independent secondary necrosis (Fig 3A). Deletion of caspase-1 in *Gsdmd*-deficient BMDMs strongly reduced LDH release, caspase-3 processing, and caspase-3 activity (Fig 3A–C). LDH levels after 3 and 5 h of poly(dA:dT) transfection were comparable with *Casp1*$^{-/-}$/*Casp11*$^{-/-}$ BMDMs, but not as low as in *Asc*$^{-/-}$, confirming that *Casp1* deletion did not affect the cell death that is caused through the ASC-Caspase-8 axis (Fig 3B). It would theoretically be possible that GSDMD-independent secondary necrosis is not driven by the catalytic activity of caspase-1, but by the formation of a caspase-1–containing scaffold and the assembly of an unknown death inducing complex, in analogy to the scaffolding function of caspase-8 (Henry & Martin, 2017). However, we found that poly(dA:dT)–induced PI influx in BMDMs from *Casp1*$^{C284A/C284A}$ mice, which express a catalytically dead caspase-1, was comparable with *Casp1*$^{-/-}$/*Casp11*$^{-/-}$ or *Casp1*$^{-/-}$ BMDMs, and much lower than PI influx in *Gsdmd*$^{-/-}$. We formally excluded this possibility (Fig 3D) and thus conclude that caspase-1 enzymatic activity is required to drive GSDMD-independent secondary necrosis.

## Bid cleavage is required for mitochondrial damage and GSDMD-independent secondary necrosis

While examining the morphology of inflammasome-activated BMDMs by confocal microscopy, we found that *Gsdmd*$^{-/-}$ cells were characterized by mitochondrial fragmentation and loss of mitochondrial membrane potential (Fig S6A) and a rapid drop of cellular ATP levels (Figs 4A and S6B–D) as early as 30 min after inflammasome activation. Given this rapid loss of mitochondrial integrity, we hypothesized that it was linked to the rapid onset of caspase-3 activation and induction of secondary necrosis in *Gsdmd*-deficient cells.

An imbalance of pro- and anti-apoptotic Bcl2 family members results in the activation of Bax/Bak pore formation and loss of mitochondrial integrity during apoptosis. Often, degradation and/or cleavage of anti-apoptotic Bcl2 proteins as well as activating cleavage of BH3-only protein are responsible for MOMP. To identify which pro-apoptotic Bcl2 proteins are processed in *Gsdmd*$^{-/-}$ BMDMs, we made use of Stable Isotope Labeling with Amino acids in Cell culture (SILAC) mass spectrometry approach (Ong et al, 2002). Differentially isotope-labelled immortalized *Gsdmd*$^{-/-}$ and *Asc*$^{-/-}$ BMDMs were transfected with poly(dA:dT), proteins separated by molecular weight using SDS–PAGE, cut according to MW and each slice analysed by mass spectrometry (Slice-SILAC). The differential analysis of the heavy versus light fraction enabled a comparison between the nonresponsive *Asc*$^{-/-}$ and the responsive *Gsdmd*$^{-/-}$, wherein appearance of smaller fragments in *Gsdmd*$^{-/-}$ indicated

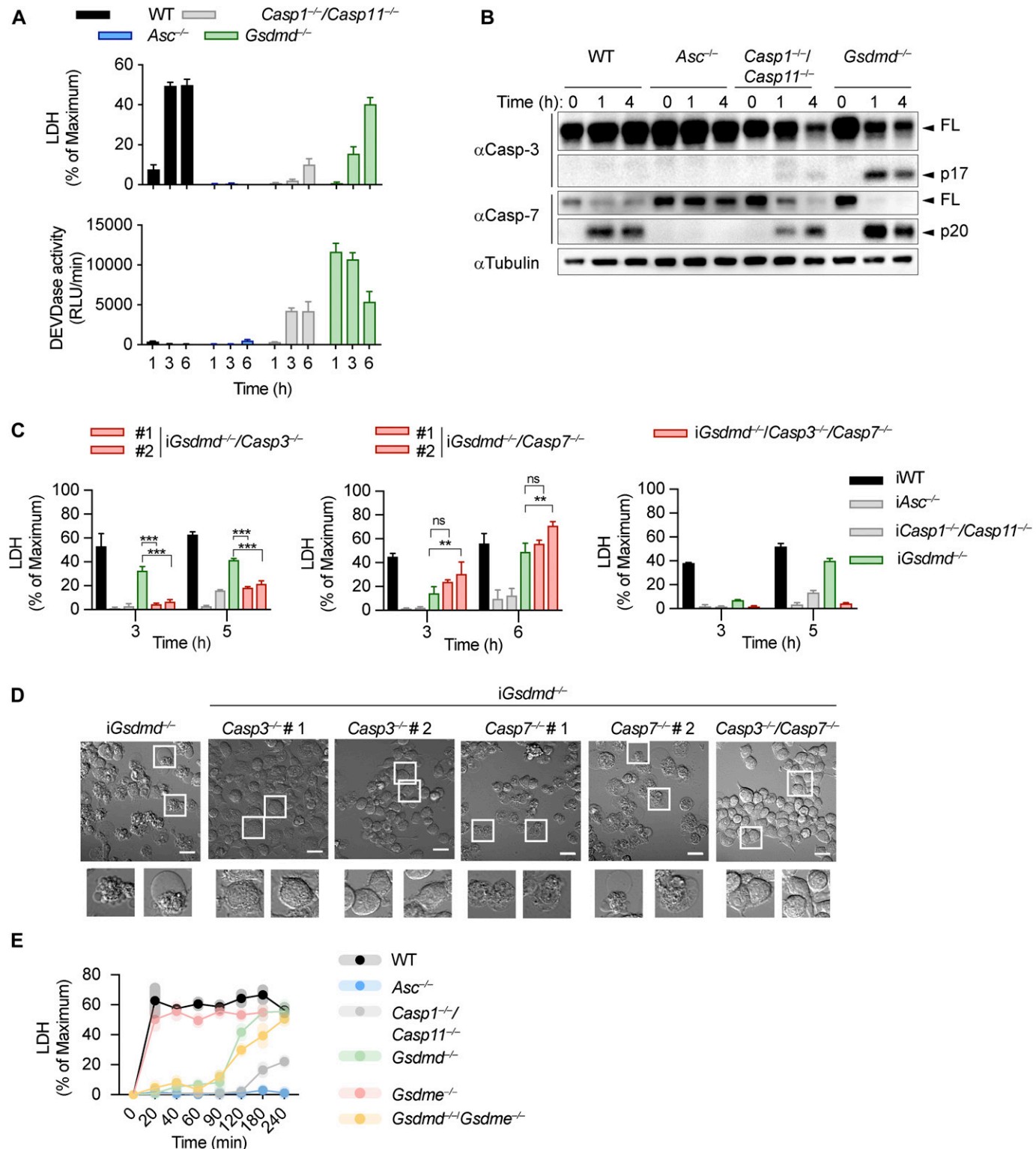

**Figure 2. Caspase-3 drives GSDMD-independent secondary necrosis in inflammasome activated cells.**
**(A, B)** LDH release, caspase-3/-7 activity (DEVDase activity) and immunoblots showing caspase-3/-7 processing from LPS-primed WT, *Asc*⁻/⁻, *Casp1*⁻/⁻/*Casp11*⁻/⁻, and *Gsdmd*⁻/⁻ primary BMDMs after transfection of poly(dA:dT). **(C)** LDH release from LPS-primed WT, *Asc*⁻/⁻, *Casp1*⁻/⁻/*Casp11*⁻/⁻, *Gsdmd*⁻/⁻, *Gsdmd*⁻/⁻/*Casp3*⁻/⁻, *Gsdmd*⁻/⁻/*Casp7*⁻/⁻, and *Gsdmd*⁻/⁻/*Casp3*⁻/⁻/*Casp7*⁻/⁻ iBMDMs after transfection of poly(dA:dT). **(C, D)** Confocal images of cells from (C). Insets show membrane ballooning in dying cells at 3 h post-transfection. Scale bar = 10 μm. **(E)** Quantification of LDH release in LPS primed WT, *Asc*⁻/⁻, *Casp-1*⁻/⁻/*Casp-11*⁻/⁻, *Gsdmd*⁻/⁻, *Gsdme*⁻/⁻, and *Gsdmd*⁻/⁻/*Gsdme*⁻/⁻ BMDMs transfected with poly(dA:dT) for 4 h. Graphs show mean ± SD. **P ≤ 0.01, ***P ≤ 0.001, "ns," no significance (unpaired *t* test). Data and blot are representative of at least three independent experiments.

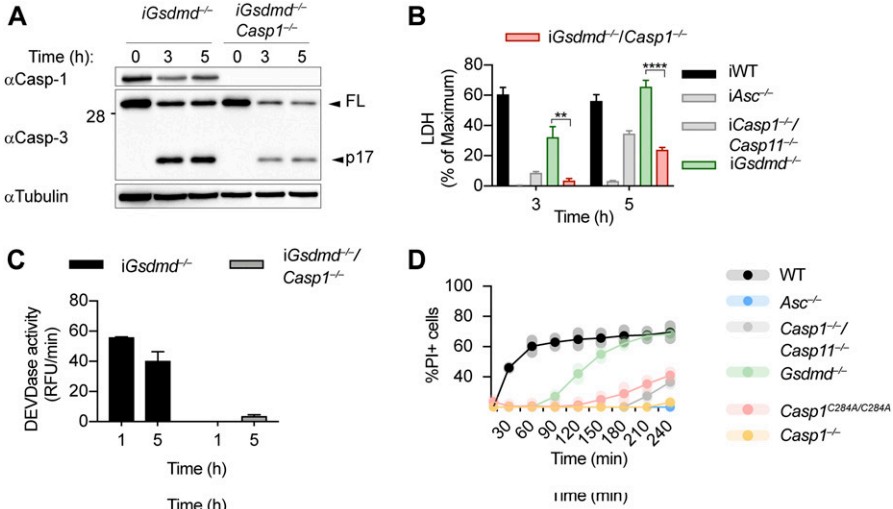

**Figure 3. Caspase-1 is required for GSDMD-independent secondary necrosis.**
**(A, B, C)** Immunoblot showing Caspase-1 expression and caspase-3 processing, LDH release, and caspase-3/-7 activity (DEVDase activity) from LPS-primed *Gsdmd*[−/−] and *Gsdmd*[−/−]/*Casp1*[−/−] immortalized BMDMs after transfection of poly(dA:dT). **(D)** PI influx of WT, *Asc*[−/−], *Casp1*[−/−]/*Casp11*[−/−], *Casp1*[−/−], *Casp1*[C284A/C284A], and *Gsdmd*[−/−] primary BMDMs after transfection of poly(dA:dT). Graphs show mean ± SD. **P ≤ 0.01, ***P ≤ 0.001, ****P ≤ 0.0001 (unpaired *t* test). Data and blot are representative of at least three independent experiments.

cleavage. We focused on potential cleavage of Bcl-2 family proteins that indicate their inability to inhibit BH3-only proteins or promote BH3-only proteins to induce MOMP (Bock & Tait, 2019). The anti-apoptotic protein Mcl-1 (of Bcl-2, Mcl-1, and Bcl-XL) and the pro-apoptotic proteins Bax, Bak, and Bid (but not Bim) were found to be cleaved in *Gsdmd*[−/−], but not in *Asc*[−/−] cells (Fig 4B). Because in type-II cells, caspase-8–cleaved tBid translocates to the mitochondria to promote Bax/Bak–dependent pore formation and intrinsic apoptosis, we investigated whether Bid cleavage promoted GSDMD-independent secondary necrosis. Confirming the SILAC data, Bid

was found to be rapidly cleaved in *Gsdmd*[−/−] cells but not in *Asc*[−/−] after inflammasome activation (Fig 4C). However, because Bid cleavage was also observed in WT and *Casp1*[−/−]/*Casp11*[−/−] BMDMs, we proceeded to assess its contribution to GSDMD-independent secondary necrosis genetically by generating *Gsdmd*[−/−]/*Bid*[−/−] iBMDMs (Fig S7A). Knocking out *Bid* in *Gsdmd*[−/−] cells significantly reduced the levels of caspase-3 activity (Fig S7B) after poly(dA:dT) transfection and in agreement with that strongly reduced LDH release and PI uptake were observed (Figs 4D and S7C). Strikingly, *Gsdmd*[−/−]/*Bid*[−/−] cells looked adhered and elongated comparable

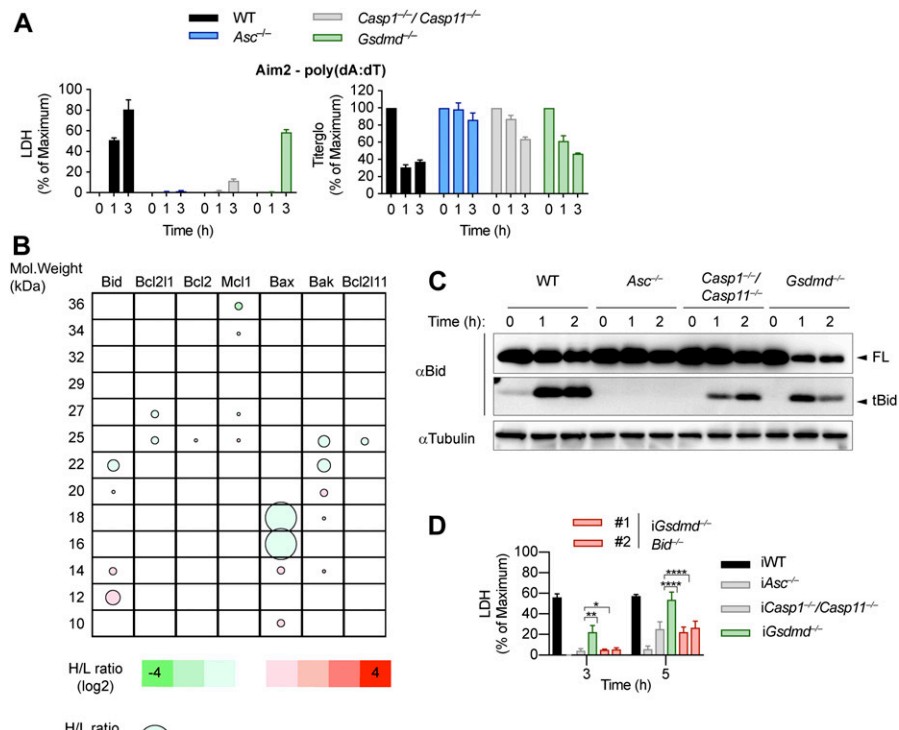

**Figure 4. Mitochondrial damage is caused by truncated Bid.**
**(A)** LDH release and Titer-Glo measurements from LPS-primed WT, *Asc*[−/−], *Casp1*[−/−]/*Casp11*[−/−], and *Gsdmd*[−/−] primary BMDMs after transfection with poly(dA:dT). **(B)** Schematic cleavage profile of Bcl-2 family members generated from slice SILAC data. Bubble diameters are proportional to the number of quantified peptide matches, whereas the gradient color represents the H/L ratio, as indicated below. The green bubbles (negative log2H/L) represent protein isoforms reduced in *Gsdmd*[−/−] iBMDMs compared with *Asc*[−/−] iBMDMs at 3 h post-poly(dA:dT) transfection; red bubbles (positive log2H/L) represent protein isoforms enriched in *Gsdmd*[−/−] iBMDMs compared with *Asc*[−/−] iBMDMs. **(C)** Immunoblots showing Bid processing from LPS-primed WT, *Asc*[−/−], *Casp1*[−/−]/*Casp11*[−/−], and *Gsdmd*[−/−] primary BMDMs after transfection with poly(dA:dT). **(D)** LDH release from WT, *Asc*[−/−], *Casp1*[−/−]/*Casp11*[−/−], *Gsdmd*[−/−], and *Gsdmd*[−/−]/*Bid*[−/−] iBMDMs after transfection with poly(dA:dT). Graphs show mean ± SD. *P ≤ 0.05, **P ≤ 0.01, ****P ≤ 0.0001, "ns," no significance (unpaired *t* test). Data and blots are representative of at least three independent experiments.

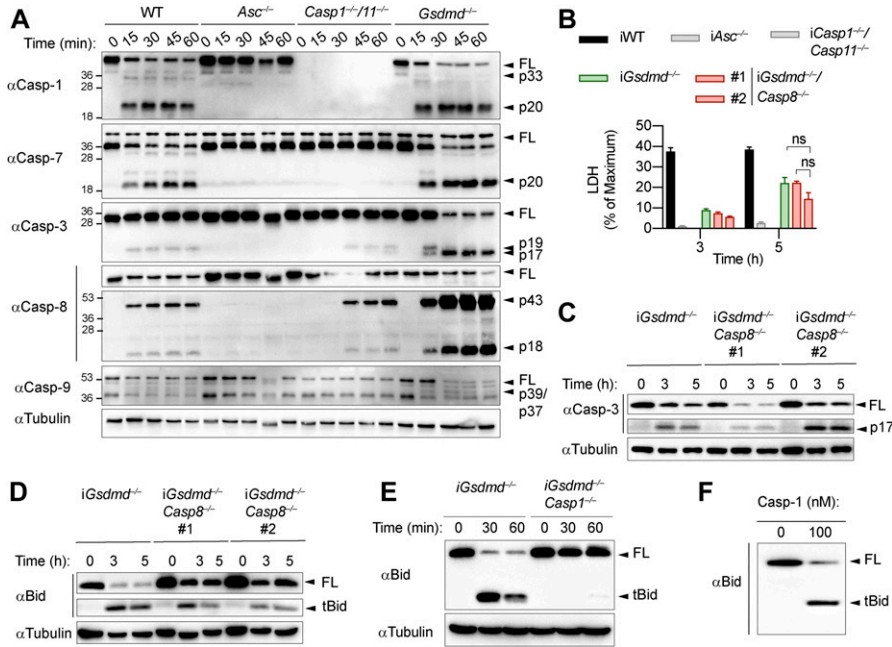

**Figure 5. Caspase-1 drives Bid processing during GSDMD-independent secondary necrosis.**
**(A)** Immunoblots showing caspase-1, caspase-7, caspase-3, caspase-8, and caspase-9 processing in WT, $Asc^{-/-}$, $Casp1^{-/-}/Casp11^{-/-}$, and $Gsdmd^{-/-}$ primary BMDMs after transfection with poly(dA:dT). **(B)** LDH release from WT, $Asc^{-/-}$, $Casp1^{-/-}/Casp11^{-/-}$, $Gsdmd^{-/-}$, and $Gsdmd^{-/-}/Casp8^{-/-}$ iBMDMs after transfection with poly(dA:dT). **(C)** Immunoblots showing caspase-3 processing in $Gsdmd^{-/-}$ and $Gsdmd^{-/-}/Casp8^{-/-}$ iBMDMs after transfection with poly(dA:dT). **(D)** Immunoblots showing Bid cleavage in $Gsdmd^{-/-}$ and $Gsdmd^{-/-}/Casp8^{-/-}$ iBMDMs after transfection with poly(dA:dT). **(E)** Immunoblots showing Bid cleavage in $Gsdmd^{-/-}$ and $Gsdmd^{-/-}/Casp1^{-/-}$ iBMDMs after transfection with poly(dA:dT). **(F)** In vitro cleavage assay showing processing of recombinant Bid by recombinant caspase-1. Graphs show mean ± SD. * "ns," no significance (unpaired t test). Data and blot are representative of at least three independent experiments.

with untreated iBMDMs upon transfection with poly(dA:dT), which is in contrast to $Gsdmd^{-/-}$ iBMDMs which displayed typical necrotic features such as rounding up, permeabilization, shrinkage, and blebbing (Fig S7D). In summary, these results show that Bid is an essential mediator of GSDMD-independent secondary necrosis and suggest that Bid cleavage is required to drive this cell death.

## Caspase-1 cleaves Bid to promote caspase-3 activation and cell lysis

Because proteolytic cleavage of Bid precedes MOMP and is required for cell death, we next enquired which upstream caspase is responsible for Bid activation. Immunoblotting for the cleaved p18 fragment of caspase-8 suggested that $Gsdmd^{-/-}$ BMDMs contain active caspase-8 at 15–30 min after poly(dA:dT) transfection, whereas very little cleaved caspase-8 p18 was found in WT, $Asc^{-/-}$, or $Casp1^{-/-}/Casp11^{-/-}$ BMDMs (Figs 5A and S8A). Interestingly, the relatively low levels of caspase-8 cleavage in $Casp1^{-/-}/Casp11^{-/-}$ compared with $Gsdmd^{-/-}$ BMDMs suggested that direct activation of caspase-8 by the ASC speck was negligible and that instead caspase-8 activation in $Gsdmd^{-/-}$ cells depended on the presence caspase-1. However, whether caspase-1 would cleave and activate caspase-8 directly or by an indirect pathway could not be deduced.

We next assessed the role of caspase-8 in causing GSDMD-independent secondary necrosis by generating $Gsdmd^{-/-}/Casp8^{-/-}$ iBMDM lines (Fig S8B). Of note, although $Casp8$ deficiency in mice results in embryonic lethality because of the unchecked activation of RIP3-dependent necroptosis (Kaiser et al, 2011; Oberst et al, 2011), $Casp8$-deficient macrophages were reported to be viable unless stimulated with extrinsic apoptotic triggers (Kang et al, 2004; Kaiser et al, 2011; Cuda et al, 2015). Indeed, when testing if $Gsdmd^{-/-}/Casp8^{-/-}$ BMDMs showed reduced levels of cell death after induction of apoptosis with the extrinsic

apoptosis stimulus TNFα/SMAC, we found that cell death was reduced, but not completely abrogated (Fig S8C). The remaining cell death, however, was block when TNFα/SMAC was combined with the RIPK3 kinase inhibitor GSK'872 (Fig S8C). These results confirmed that the cells were indeed $Casp8$ knockouts and that the necroptotic pathway was only initiated when death receptors were engaged. We next compared LDH release in $Gsdmd^{-/-}$ and $Gsdmd^{-/-}/Casp8^{-/-}$ BMDMs after transfection of the AIM2 inflammasome activator poly(dA:dT). Unexpectedly, we found no difference in LDH release nor PI uptake between these two genotypes (Figs 5B and S8D). Furthermore, we were still able to detect Bid cleavage and caspase-3 processing to the active p17 fragment in inflammasome-activated $Gsdmd^{-/-}/Casp8^{-/-}$ BMDMs (Fig 5C and D). Previous work has implied that Bid can also be a substrate of caspase-1 (Li et al, 1998) because caspase-1 and caspase-8 have partially overlapping substrate spectrum that includes also GSDMD and IL1β (Maelfait et al, 2008; Orning et al, 2018; Sarhan et al, 2018; Chen et al, 2019). In line with caspase-1 controlling Bid cleavage directly and independently of caspase-8, we found that tBid generation after AIM2 activation was completely abrogated in $Gsdmd^{-/-}/Casp1^{-/-}$ BMDMs at an early time point and strongly reduced after prolonged incubation (Fig 5E) and that caspase-1 was able to efficiently convert Bid to tBid in an in vitro cleavage assay (Fig 5F). In summary, our data thus far suggest that although Bid cleavage is essential for GSDMD-independent secondary necrosis and high levels of active caspase-8 are found in these cells, it is caspase-1 and not caspase-8 that processes Bid and induces mitochondrial permeabilization.

## GSDMD-independent secondary necrosis requires both caspase-8 and caspase-9

Having identified caspase-1, Bid and caspase-3 as the essential drivers of GSDMD-independent secondary necrosis, we next asked if activation

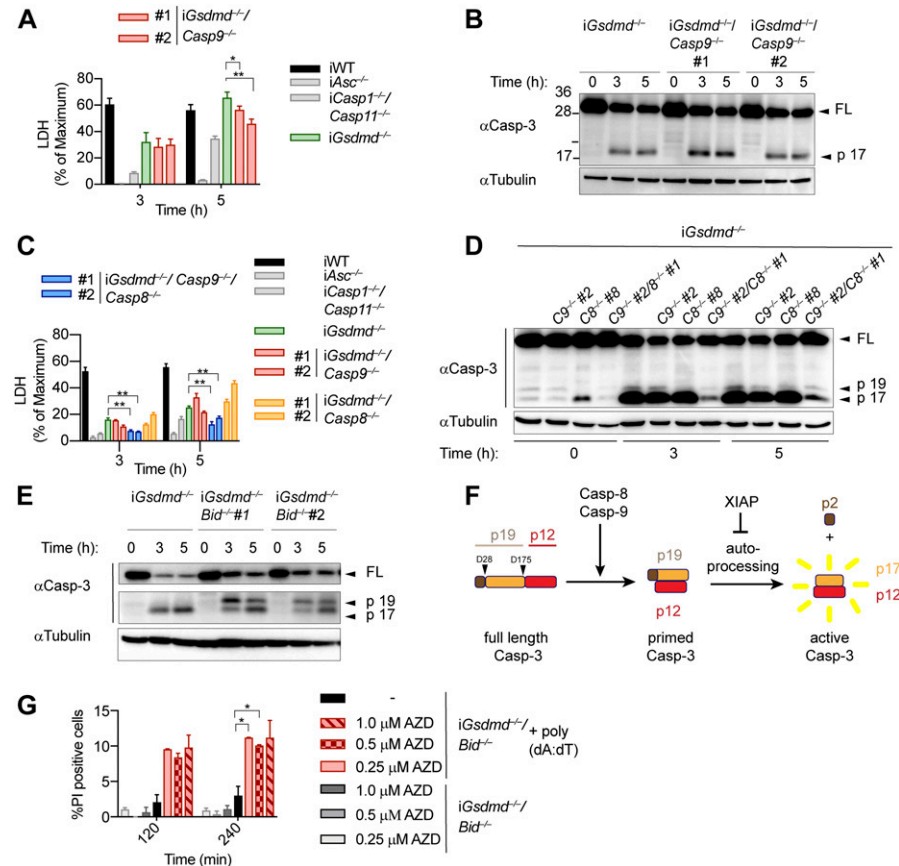

**Figure 6. SMAC release and initiator caspases-8/-9 are required for GSDMD-independent secondary necrosis.**
**(A)** LDH release from WT, $Asc^{-/-}$, $Casp1^{-/-}/Casp11^{-/-}$, $Gsdmd^{-/-}$, and $Gsdmd^{-/-}/Casp9^{-/-}$ iBMDMs after transfection with poly(dA:dT). **(B)** Immunoblots showing caspase-3 cleavage in $Gsdmd^{-/-}$ and $Gsdmd^{-/-}/Casp9^{-/-}$ iBMDMs after transfection with poly(dA:dT). **(C)** LDH release from WT, $Asc^{-/-}$, $Casp1^{-/-}/Casp11^{-/-}$, $Gsdmd^{-/-}$, $Gsdmd^{-/-}/Casp8^{-/-}$, $Gsdmd^{-/-}/Casp9^{-/-}$, and $Gsdmd^{-/-}/Casp8^{-/-}/Casp9^{-/-}$ iBMDMs after transfection with poly(dA:dT). **(D)** Immunoblots showing caspase-3 cleavage in $Gsdmd^{-/-}$, $Gsdmd^{-/-}/Casp8^{-/-}$, $Gsdmd^{-/-}/Casp9^{-/-}$, and $Gsdmd^{-/-}/Casp8^{-/-}/Casp9^{-/-}$ iBMDMs after transfection with poly(dA:dT). **(E)** Immunoblots showing caspase-3 processing from $Gsdmd^{-/-}$ and $Gsdmd^{-/-}/Bid^{-/-}$ iBMDMs after transfection with poly(dA:dT). **(F)** Schematic summary of the mechanism of caspase-3 cleavage and activation. **(G)** PI influx in untreated or poly(dA:dT)–transfected $Gsdmd^{-/-}/Bid^{-/-}$ iBMDMs in the presence or absence of the SMAC mimetic AZD5582. Graphs show mean ± SD. *$P \leq 0.05$, **$P \leq 0.01$ (unpaired $t$ test). Data and blot are representative of at least three independent experiments.

of caspase-9 downstream of mitochondrial permeabilization and cytochrome c release provides the link between Bid and caspase-3 activation. We, thus, generated $Gsdmd^{-/-}/Casp9^{-/-}$ iBMDM lines by CRISPR/Cas9 genome targeting and verified that they lacked caspase-9 expression and no longer responded to intrinsic apoptosis induction (Fig S9A and B).

However, we found that in analogy to $Casp8$-deficiency, knocking out of $Casp9$ in $Gsdmd^{-/-}$ had only a small impact on poly(dA:dT)–induced secondary necrosis after 5 h of treatment, whereas no impact was detectable at an earlier time point (Figs 6A and S9C). Furthermore, caspase-3 processing was also found to be unaffected in these cell lines (Fig 6B).

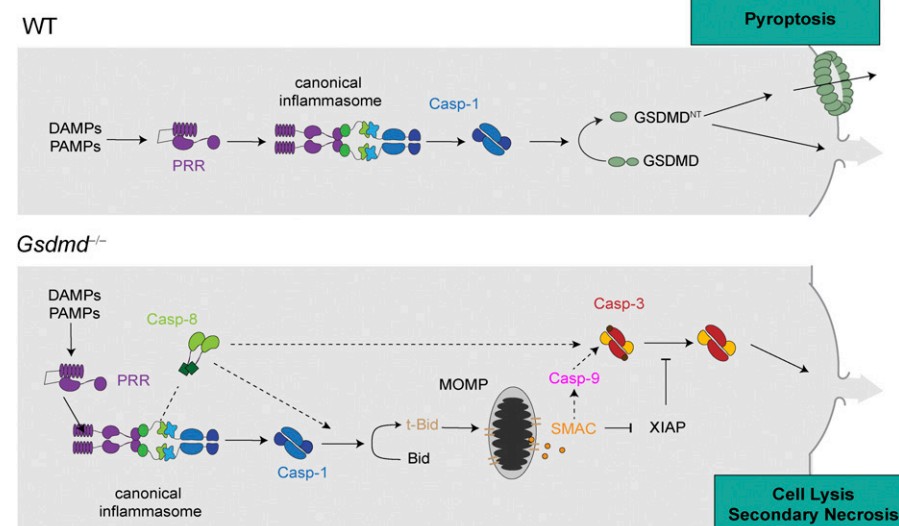

**Figure 7. Model of cell death in $Gsdmd$-deficient myeloid cells after activation of caspase-1.**
Model depicting the mechanism of canonical inflammasome activation in WT cells undergoing caspase-1– and GSDMD-dependent pyroptosis and $Gsdmd^{-/-}$ cells undergoing caspase-1–induced GSDMD-independent secondary necrosis.

These results raised the possibility that caspase-8 and caspase-9 activity was redundant or that caspase-1 was driving Bid cleavage and caspase-3 activation somehow independently of both initiator caspases. We addressed these two scenarios by creating $Gsdmd^{-/-}$/$Casp8^{-/-}$/$Casp9^{-/-}$ iBMDM lines (Fig S9D) and compared their phenotype after AIM2 inflammasome activation to our other knockout lines. Poly(dA:dT)–transfected $Gsdmd^{-/-}$/$Casp8^{-/-}$/$Casp9^{-/-}$ BMDMs displayed significantly reduced levels of LDH release compared with $Gsdmd^{-/-}$, $Gsdmd^{-/-}$/$Casp8^{-/-}$, or $Gsdmd^{-/-}$/$Casp9^{-/-}$ cells (Fig 6C). Consistent with the reduced levels of cell lysis, we also found that caspase-3 processing was significantly reduced in $Gsdmd^{-/-}$/$Casp8^{-/-}$/$Casp9^{-/-}$ when compared with the other genotypes (Fig 6D), confirming that activity of either initiator caspase was sufficient to drive caspase-3 activation and GSDMD-independent secondary necrosis.

**Bid-induced mitochondrial permeabilization is required to release SMAC and promote conversion of caspase-3 p19 to p17**

The finding that single deficiency in either caspase-8 or caspase-9 had no impact on caspase-3 activation and GSDMD-independent secondary necrosis, whereas double-deficiency abrogated cell lysis was unexpected and puzzling. Because Bid was essential for GSDMD-independent secondary necrosis, whereas caspase-9 was not, we hypothesized that other factors released from permeabilized mitochondria were required. Besides cytochrome c, which activates Apaf-1 to assemble the apoptosome and promote caspase-9 activity, mitochondria also release ATP and SMAC. SMAC binds IAPs, in particular XIAP, which normally suppresses caspase-3/7 and caspase-9 activity, and thus relieves the block on apoptosis induction (Deveraux et al, 1997; Wu et al, 2000). We, thus, closely examined caspase-3 processing between poly(dA:dT)–transfected $Gsdmd^{-/-}$ and $Gsdmd^{-/-}$/$Bid^{-/-}$ and found that while only the p17 fragment of caspase-3 was found in $Gsdmd^{-/-}$ BMDMs, $Gsdmd^{-/-}$/$Bid^{-/-}$ featured two cleaved caspase-3 bands, at 19 and 17 kD (Figs 6E and S9E). Previous studies showed that p19 fragment is generated by apical caspases cleaving in the linker domain between the large and small subunit, whereas the p17 is generated by auto-processing of the pro-peptide by caspase-3 itself (Kavanagh et al, 2014) (Fig 6F). We hypothesize that Bid deficiency delayed IAP release and, thus, conversion from the p19 to the p17 fragment and full activity of caspase-3 and that this was a critical factor for GSDMD-independent secondary necrosis. Indeed, treatment with the SMAC mimetic AZD5582 increased generation of caspase-3 p17 (Fig S9F) and partially restored cell death in $Gsdmd^{-/-}$/$Bid^{-/-}$ BMDMs (Fig 6G). These results suggest that during GSDMD-independent secondary necrosis, Bid cleavage and mitochondrial permeabilization are mainly required for the release of SMAC and subsequent binding to XIAP but not to drive caspase-9 activation. However, because either caspase-8 or caspase-9 are needed to process caspase-3 (Fig 6D), caspase-1 cannot induce GSDMD-independent secondary necrosis in the absence of both initiator caspases (Fig 7).

## Discussion

Here, we show that the cell lysis that occurs in $Gsdmd$-deficient cells upon activation of canonical inflammasomes is a rapid form of secondary necrosis (referred to as "GSDMD-independent secondary necrosis" in this article) and that it depends on the caspase-1–dependent activation of either caspase-8 or caspase-9, Bid cleavage, SMAC release, and caspase-3 activity. Secondary necrosis describes the loss of membrane integrity of apoptotic cells or apoptotic bodies and is, thus, appropriate because death in $Gsdmd$-deficient cells relies on initiator caspases and the executor caspase-3 and results in a loss of membrane integrity. Yet, it is remarkably different from regular apoptosis in the signalling pathways that underlie its induction, cellular morphology, and the speed by which cells undergo death.

GSDMD-independent cell lysis is characterized by a rapid loss of mitochondrial potential and rapid activation of caspase-3 and an atypical apoptotic morphology. Indeed, cells undergoing this type of cell death show only initially the signs of regular apoptotic blebs or apoptotic body formation and quickly lose membrane integrity and start ballooning, similarly to pyroptotic cells. This morphology has in the past led to the speculation that caspase-1 might directly or indirectly cleave an alternative lytic cell death executor, such as another gasdermin family member. Indeed, recently Tsuchiya et al (2019) proposed that caspase-3 processes GSDME in $Gsdmd$-deficient CL26 cells that harbor dimerizer-activated caspase-1 (Tsuchiya et al, 2019). Our data in primary mouse macrophages, however, show no involvement of GSDME in GSDMD-independent secondary necrosis (Figs 2E and S5A and B), although GSDME is detectable and processed (Fig S5B). This discrepancy is most likely caused by differences in GSDME expression levels between different cell types. Recent results show that a number of cancer cell lines express sufficiently high levels of GSDME to cause pyroptosis upon treatment with apoptosis-inducing chemotherapy drugs (Wang et al, 2017b). It remains to be determined how much GSDME is expressed by CL-26 cells, a murine colorectal carcinoma cell line, compared with macrophages (Tsuchiya et al, 2019), but overwhelming evidence suggest that at least in macrophages, GSDME expression or activity appear to be insufficient to induce GSDME-dependent cell death after caspase-3 activation (Lee et al, 2018; Sarhan et al, 2018; Vince et al, 2018; Chen et al, 2019). Thus, other yet undefined factors drive the lysis of $Gsdmd$-deficient BMDMs.

Another striking difference between regular apoptosis and GSDMD-independent secondary necrosis is the signalling pathway underlying caspase-3 activation. Our data show that the main driver of this cell death is active caspase-1 and that it promotes cell death by cleaving several targets. The most critical of these targets appears to be Bid, which is converted by caspase-1 to tBid (independently of caspase-8) and which induces mitochondrial permeabilization and the release of cytochrome c, ATP, and SMAC. Moreover, caspase-1 acts as a kind of "super-initiator" caspase by activating initiator caspases-8/-9. It is worth noting that caspase-8 activation is mostly driven by caspase-1 with a negligible contribution of direct caspase-8 activation at the ASC speck, as evident from much reduced caspase-8 cleavage in $Casp1$/$Casp11$–deficient compared with $Gsdmd$-deficient cells. This could potentially be driven by direct caspase-1–induced cleavage of caspase-8, or by caspase-1, somehow enhancing ASC-dependent caspase-8 activation. Caspase-9 activation, however, is downstream of Bid cleavage. The requirement for either caspase-8 and casoase-9

appears to stem from the fact that caspase-1 fails to process caspase-3 efficiently, despite previous reports suggesting that caspase-1 cleaved caspase-3 directly (Taabazuing et al, 2017; Sagulenko et al, 2018). However, caspase-1 is efficient enough to activate Bid to induce SMAC release, to relieve inhibition by IAPs, and allows full conversion to caspase-3 p17/p12.

Our findings are in contradiction to the recent report by the Suda Laboratory, which proposed that cell death in *Gsdmd*-deficient cells is solely caused by the Bid–caspase-9–caspase-3 axis (Tsuchiya et al, 2019). The discrepancy is potentially related to cell line–intrinsic differences or to the method used to activate caspase-1. Tsuchiya et al (2019) performed experiments in CL26 cells, which, for example, lack ASC and, thus, lack the ASC speck-induced activation of caspase-8 (Pierini et al, 2012; Sagulenko et al, 2013; Vajjhala et al, 2015; Fu et al, 2016), whereas we used immortalized macrophages, which recapitulate the behavior of primary BMDMs. Furthermore, they used a dimerizer-based system to activate caspase-1, which most likely induces higher levels of caspase-1 activity compared with physiological inflammasome triggers and, thus, might explain why Tsuchiya et al (2019) did not observe a role for caspase-8, which we find necessary to amplify caspase-1 activity after treatment with canonical inflammasome triggers. However, both studies agree that Bid cleavage is essential for cell death in *Gsdmd*-deficient cells and that Bid is cleaved by caspase-1 independently of caspase-8.

Recent work has revealed a surprisingly high level of redundancy and cross talk between the apoptotic, necroptotic, and pyroptotic cell death pathways. Interestingly, in many cases, these pathways or the cross talk are normally not detectable or only turned on when another pathway is inhibited. For example, deletion of caspase-8 or mutation of its auto-processing sites are known to result in activation of RIP3/MLKL-dependent necroptosis, a pathway that is otherwise not observed, and catalytic-dead caspase-8 results in activation of necroptosis and pyroptosis (Kaiser et al, 2011; Oberst et al, 2011; Kang et al, 2018, Newton, 2019a; 2019b). It is assumed that this redundancy developed as a defense mechanism to guard against pathogen-induced inhibition of apoptosis, and accordingly viral inhibitors of the three major cell death pathways have been identified (Li & Stollar, 2004; Taxman et al, 2010; Nailwal & Chan, 2019), which highlights that necroptosis is not an artifact caused by lack of caspase-8 activity. Similarly, it could be speculated that the ability of caspase-1 to induce rapid secondary necrosis by activating apoptotic caspases might have developed as a safeguard against viruses that inhibit GSDMD. Indeed, recently, the pathogenic enterovirus 71, which is known to trigger the NLRP3 inflammasome (Wang et al, 2017a), was shown to interfere with GSDMD activation. In particular, the viral protease 3C was shown to cleave GSDMD at Q193/194, interfering with N-terminal fragment formation, oligomerization, and GSDMD pore formation (Wang et al, 2015). Furthermore, GSDMD-independent secondary necrosis appears to contribute to the clearance of bacterial infection, as it could be shown that *Gsdmd*$^{-/-}$ mice are less susceptible to infection with *Francisella novicida* compared with *Casp1*- or *Aim2*-deficient animals (Schneider et al, 2017; Kanneganti et al, 2018b). Along the same lines, *Gsdmd*-deficient mice infected with *Burkholderia thailandensis* show lower CFUs and lower IL-1β levels than *Casp1/Casp11*–deficient animals (Wang et al, 2019). Similarly, it was reported that peritoneal IL-1β levels are higher in *Salmonella typhimurium*–infected *Gsdmd*$^{-/-}$ mice than *Casp1*$^{-/-}$ controls (Monteleone et al, 2018). These studies, thus, allow the conclusion that GSDMD-independent cell death is also engaged in vivo and that it allows partial protection against intracellular bacterial pathogens. Unexpectedly, however, GSDMD-independent secondary necrosis does not appear to be important in models of autoinflammatory diseases because *Gsdmd* deficiency rescues mice expressing mutant NLRP3 or Pyrin, linked to neonatal-onset multisystem inflammatory disease and familial Mediterranean fever (Xiao et al, 2018; Kanneganti et al, 2018a).

Considering that knockout of GSDMD showed a big improvement in pro-inflammatory symptoms associated with the autoinflammatory diseases neonatal-onset multisystem inflammatory disease and familial Mediterranean fever and the importance of the canonical inflammasome pathway in sterile inflammatory disease, research has focused on the discovery of GSDMD-specific inhibitors. To date, several inhibitors have been identified, although off-target effects and specificity still need to be evaluated in more detail (Rathkey et al, 2018; Sollberger et al, 2018; Rashidi et al, 2019). Furthermore, it is important to consider that caspase-1 activity is unrestrained by these inhibitors and that, thus, caspase-1 might induce cell death and inflammation through the back-up pathway described in our study.

# Materials and Methods

## Antibodies, chemicals, and reagents

### Drugs
VX-765 (MedChemExpress), Caspase-3/7 inhibitor I (CAS 220509-74-0; Santa Cruz Biotechnology), Q-VD-Oph (Selleck Chemicals), AZD5582 (Selleck Chemicals), 7-Cl-O-Nec1 (Abcam), GSK872 (Selleck Chemicals), K777 (AdipoGen), PD 150606 (Tocris), Calpeptin (Selleck Chemicals), ABT-737 (Selleck Chemicals), S63845 (Selleck Chemicals), and Nigericin (InvivoGen).

### Antibodies
GSDMD (Ab209845; Abcam), Casp-1 (Casper1, AG-20B-0042-C100; AdipoGen), Tubulin (Ab40742; Abcam), IL-1β (AF-401-NA; R&D Systems), Caspase-3 (#9662; Cell Signaling Technology), Caspase-7 (#9492; Cell Signaling Technology), Caspase-8 (#9429 and 4927; Cell Signaling Technology), Caspase-9 (#9508 and #9504; Cell Signaling Technology), and Bid (#2003; Cell Signaling Technology).

## Animal experiments

All experiments were performed with approval from the veterinary office of the Canton de Vaud and according to the guidelines from the Swiss animal protection law (license VD3257). C57BL/6J mice were purchased from Janvier Labs and housed at specific pathogen-free facility at the University of Lausanne. Mice lacking *Asc*, *Casp1*, *Casp1/11*, *Gsdmd*, *Gsdme*, or expressing mutant *Casp1*$^{C284A}$ have been previously described (Mariathasan et al, 2004; Kayagaki et al, 2011; Schneider et al, 2017; Chen et al, 2019). All mice were either generated (*Gsdmd*$^{-/-}$ and *Gsdme*$^{-/-}$) or backcrossed (other lines) in the C56BL/6J background.

## Cell culture and immortalization of macrophages

Primary mouse macrophages (BMDMs) were differentiated for 6 d and cultured for up to 9 d in DMEM (Gibco) supplemented with 10% FCS (Bioconcept), 20% 3T3 supernatant (MCSF), 10% Hepes (Gibco), and 10% nonessential amino acids (Gibco). Immortalization of macrophages was performed as previously described (Blasi et al, 1985; Broz et al, 2010). Immortalized macrophages (iBMDMs) were cultured in DMEM complemented with 10% FCS (Bioconcept), 10% MCSF (3T3 supernatant), 10% Hepes (Amimed), and 10% nonessential amino acids (Life Technologies). To passage the BMDMs and iBMDMs, the cells were washed with PBS and left to detach at 4°C for 15 min and scarped using cell scrapers (Sarstedt), spun down at 300$g$ for 5 min at 4°C, and resuspended in the appropriate amount of medium.

## Crispr genome editing in immortalized macrophages

*Bid*-, *Casp9*-, *Casp-8*-, *Casp-8/Casp9*-, *Casp1*-, *Casp3*-, *Casp7*-, and *Casp3/7*-deficient iBMDMs were generated using the genome-editing system Alt-R-CRISPR/Cas (IDT) according to the manufacturer's protocol. Briefly, the gene-specific targeting crRNA (Bid: TGGCTGTACTCGCCAAGAGC TGG Caspase-9: CACACGCACGGGCTCCAACT TGG, Caspase-8: CTTCCTAGACTGCAACCGAG AGG, Caspase-1: AAT-GAAGACTGCTACCTGGC AGG, Caspase-7: GATAAG TGGGCACTCGGTCC TGG, and Caspase-3: AATGTCATCTCGCTCTGGTA CGG or TGGGCC-TGAAATACCAAGTC AGG) was mixed with the universal RNA oligo tracrRNA to form a gRNA complex (crRNA–tracrRNA). The addition of the recombinant Cas9 nuclease V3 allowed the formation of an RNP complex specific for targeting the desired genes. The tracrRNA only or RNP complexes were subsequently reverse transfected into either WT or *Gsdmd*$^{-/-}$ immortalized iBMDMs using RNAiMax (Invitrogen). The bulk population was tested for successful gene mutation using the T7 endonuclease digestion assay as follows: the cells were lysed by the KAPA Biosystems Kit according to the manufacturer's protocol, and genomic DNA flanking the guide RNA (crRNA)–binding site was amplified by PCR using gene-specific primers (Bid: fw: CTGGACATTACTGGGGGCAG, rv: CTCGATAGCCC-CTTGGTGTC; Caspase-9: fw: CAAGCTCTCCAGACCTGACC, rv: GAGATCT-GACGGGCACCATT; Caspase-8: fw: GGGATGTTGGAGGAAGGCAA, rv: GGCACAGACTTTGAGGGGTT; Caspase-1: fw: CAGACAAGATCCTGAGGGCA, rv: AGATGAGGATCCAGCGAGTAT; Caspase-7: fw: TTGCCTGACCCAAG GTTTGT, rv: CCCAGCAACAGGAAAGCAAC; and Caspase-3: fw: GTG GGGGA-TATCGCTGTCAT, rv: TGTGTAAGGATGCGGACTGC). The amplified genomic DNA was used to perform the heteroduplex analysis according to the manufacturer's protocol (IDT). Single clones were derived from the bulk population by limiting dilution, and the absence of protein expression in single clones was verified by immunoblotting and sequencing of genomic regions, where required.

## siRNA knockdown

$2.5 \times 10^5$ *Gsdmd*-deficient iBMDMs were seeded per well of a six-well plate and incubated overnight. For the siRNA transfection, the medium was changed to OptiMEM, and siRNA transfection was carried out according to the manufacturer's protocol, transfecting 25 pmol siRNA (non-targeting: siGENOME non-targeting siRNA control pools [D-001206-14; Dharmacon], caspase-3: Casp3 SMART POOL [M-043042-01; Dharmacon], and caspase-7: Casp7 SMART POOL [M-057362-01; Dharmacon]) with 7.5 $\mu$l Lipofectamine RNAiMax (Invitrogen) per well. Medium was exchanged for DMEM (10% FCS, 10% MCSF, 1% NeAA, and 1% Hepes) after 6 h. 48 h post-transfection, the cells were collected and reseeded in a 96-well plate at $3 \times 10^4$ cells/well. The cells were primed and treated as in cell death assays.

## Cell death assays

The cells were seeded in 96-well plates (100 $\mu$l/well) or 12-well plates (1 ml/well) at a density of $0.5 \times 10^6$ cells/ml overnight and primed the next day with 100 ng/ml ultrapure LPS-B5 (055:B5; InvivoGen) for 4 h. AIM2 inflammasome activation was achieved by transfecting 0.4 $\mu$g poly(dA:dT) (InvivoGen) per $10^5$ cells. In separate tubes, poly(dA:dT) and linear polyethylenimine (1 $\mu$g per $10^5$ cells; PolyScience) were mixed with OptiMEM by vortexing and left for 3 min at room temperature. Then poly(dA:dT) and poly-ethylenimine (PEI) were mixed together, vortexed shortly, and left for 15 min before adding a quarter of the total volume on top of the cells. Transfection was facilitated by spinning cells for 5 min at 300$g$ at 37°C. *Salmonella enterica* serovar Typhimurium SL1344 and *Francisella tularensis* subsp. *novicida* U112 (*F. novicida*) infection were performed in OptiMEM. For *S. typhimurium* infection, bacteria were grown overnight and subcultured 1/40 for 3 h and 30 min in Luria low-salt broth (LB low salt) supplemented with appropriate antibiotics, whereas infection with *Francisella* were performed from the overnight culture grown in brain heart infusion broth supplemented with 0.2% L-cysteine (Sigma-Aldrich) and appropriate antibiotics. Bacteria were then added on top of the cells in OptiMEM, spun at 300$g$ for 5 min and incubated at 37°C for the duration of the experiment or extracellular bacterial growth suppressed by addition of gentamycin at 30 and 120 min postinfection for *S. typhimurium*, and *F. novicida*, respectively. For the NLRP3 inflammasome activation, LPS-B5 (055:B5; InvivoGen) priming was carried out in OptiMEM for 4 h before addition of 5 $\mu$M nigericin (Sigma-Aldrich) and incubated for indicated time. Similarly, the cells were primed with 100 ng/ml LPS LPS-B5 (055:B5; InvivoGen) for 4 h in OptiMEM. LPS/FuGeneHD complexes were prepared by mixing 100 $\mu$l OptiMEM with 2 $\mu$g ultrapure LPS O111:B4 (InvivoGen) and 0.5 $\mu$l of FuGENE HD (Sigma-Aldrich) per well to be transfected. The transfection mixture was vortexed briefly, incubated for 10 min at room temperature, and added dropwise to the cells. The plates were centrifuged for 5 min at 200$g$ and 37°C. Extrinsic apoptosis was induced by adding 100 ng/ml TNF-$\alpha$ and the SMAC mimetic AZD5582 at the indicated concentration. Intrinsic apoptosis was induced by addition of the BH3 mimetic small molecule inhibitor ABT-737 in combination with the Mcl-1 inhibitor S63845 at the indicated concentrations.

## Cell death and cytokine release measurement

Cell lysis was assessed by quantifying the amount of lactate dehydrogenase in the cell supernatant using the LDH cytotoxicity kit (Takara) according to the manufacturer's instructions. To measure cell permeabilization, propidium iodide (Thermo Fisher Scientific) was added to the medium at 12.5 $\mu$g/ml and fluorescent

emission measured by Cytation5 (Biotek) over time. LDH and PI uptake were normalized to untreated control and 100% lysis. Cytokine release into the supernatant in particular IL-1β was measured by Elisa (Thermo Fisher Scientific) according to the manufacturer's instructions.

## DNA fragmentation assay

DNA fragmentation during apoptosis and pyroptosis was assessed by agarose gel electrophoresis as described before (Kasibhatla et al, 2006). In brief, *Gsdmd*$^{-/-}$ BMDMs were seeded in a 12-well plate and treated with apoptotic triggers or transfected with poly(dA:dT) as described under Cell Death Assays section.

## Cell lysis and immunoblotting

After treatment of cells, cell supernatant was collected and 1× sample buffer (Thermo Fisher Scientific) complemented with 66 nM Tris, and 2% SDS was added to the cell lysate. The proteins of the supernatant were precipitated on ice using an end concentration of 4% TCA (wt/vol) for 30 min. Supernatant was then spun down at 20,000$g$ for 20 min at 4°C washed with 100% acetone and centrifuged at 20,000$g$ for 20 min at 4°C. The protein pellet was air-dried and resuspended with the lysate. The samples were boiled for 10 min at 70°C and separated by a 10% or 12% SDS page gel. Transfer to the 0.2 $\mu$M Polyvinylidene fluoride (PVDF) membranes was accomplished by Trans-Blot Turbo System. The membranes were blocked with 5% milk in TBS-T and incubates with the primary antibody for 2 h at RT or overnight. Membranes were washed three times with TBS-T and HRP-coupled antibodies added in 5% milk in TBST-T for 1 h. After washing, the membranes were revealed by FUSION imager (VILBER) using Pierce ECL Western Blotting Substrate (Thermo Fisher Scientific) or Pierce ECL Plus Western Blotting Substrate (Thermo Fisher Scientific).

## Live cell imaging

BMDMs or iBMDMs were seeded 5 × 10$^4$/well in eight-well tissue culture–treated $\mu$-Slides (iBidi) or 96-well Cell Culture Microplates, $\mu$Clear (Greiner Bio-One) overnight and primed the next day with 100 ng/ml LPS 055:B5 for 4 h. The AIM2 inflammasome was activated by transfection of poy(dA:dT) (see the Cell Death Assay section). For time-lapse microscopy, cells were incubated with CellTox Green (Promega) 1:10,000 and AnnexinV (BioLegend) at 500 ng/ml or for mitochondrial health assessment MitoTracker Green and MitoTracker CMXRos were added to OptiMEM at a final concentration of 125 nM. Images were taken every 5 min or every 15 min, respectively. Zeiss LSM800 point scanning confocal microscope equipped with 63× Plan-Apochromat NA 1.4 oil objective, Zeiss ESID detector module, LabTek heating/CO$_2$ chamber, and motorized scanning stage.

## Slice-SILAC

*Gsdmd-* and *Asc*-deficient iBMDMs were grown in SILAC DMEM (Thermo Fisher Scientific) medium supplemented with 10% dialyzed FBS, 200 mg/ml proline, 150 mg/ml heavy or light lysine and 50 mg/ml arginine, respectively. The cells were passaged five to six times until 100% labelling was achieved. For the experiment, the cells were seeded at 5 × 10$^5$/well in 12-well plates overnight and primed the next day with 100 ng/ml LPS 055:B5 for 4 h. Poly(dA:dT) transfection was then carried out as described under Cell Death Assays section and plates incubated for 3 h. The cells were scraped in OptiMEM, and proteins were precipitated by 4% TCA. The obtained protein pellet was then resuspended in FASP buffer (4% SDS, 0.1 M DTT, and 100 mM Tris, pH 7.5), heated for 5 min at 95°C, sonicated, and cleared by 10-min centrifugation at 12,000$g$. Downstream sample preparation, including SDS gel preparation, mass spectrometry, and data analysis have been described before (Di Micco et al, 2016).

## Caspase activity assay

Caspase-3/7 activity was either measured by luminescence using the Caspase-Glo 3/7 (Promega) according to the manufacturer's protocol or by fluorescence. The caspase activity assay was performed as follows using the fluorescent substrate N-acetyl-Asp-Glu-Val-Asp-7-amido-4-trifluoromethylcoumarin (Sigma-Aldrich). The cells were lysed directly in the medium by adding 5× lysis buffer (250 mM Hepes, 25 mM CHAPS, and 25 mM DTT) and pipetting up and down. 30 $\mu$l of lysed cells was incubated with 30 $\mu$l of 2× assay buffer (40 mM Hepes, 200 mM NaCl, 2 mM EDTA, 0.2% CHAPS, 20% sucrose, and 20 mM DTT), and 50 $\mu$M final concentration of substrate was taken in black opaque OptiPlate-96 (PerkinElmer) and read at 400/505 at 37°C every 2 min for 10 min.

## Metabolic activity—ATP content

Metabolic activity was measured by Titer-Glo (Promega) according to the manufacturer's protocol. In brief, cells plus 25 $\mu$l supernatant were incubated with 25 $\mu$l Titer-Glo, shook for 2 min at 600 rpm, and incubated for 10 min at room temperature before reading.

## In vitro caspase cleavage assay

Active recombinant caspase-1 was purified as described before (Sborgi et al, 2016). For the in vitro cleavage assay, cell lysate from i*Gsdmd*$^{-/-}$/*Casp-3*$^{-/-}$/*Casp-7*$^{-/-}$ was prepared as described before (Boucher et al, 2012). Briefly, the cells were lysed in ice-cold buffer (50 mM Hepes, pH 7.4, 150 mM NaCl, and 1% IGEPAL) and incubated on ice for 30 min. Cellular proteins were recovered by centrifugation at 7,000$g$ for 10 min and kept at −80°C in 30-$\mu$l aliquots. Purified active caspase-1 (50, 100 nM) was added to cell lysate and incubated for 2 h at 37°C. The mixture was then analysed by immunoblot.

# Supplementary Information

# Acknowledgements

This work was supported by grants from the European Research Council (ERC-2017-CoG—770988—InflamCellDeath) and from the Swiss National Science Foundation (310030_175576) to P Broz. KW Chen is supported by a Marie Skłodowska-Curie Actions (MSCA) incoming fellowship (MSCA-IF-2018-838252). We thank Prof Dr Thomas Henry and Prof Dr Olaf Gross for sharing *Asc*$^{-/-}$, *Casp1*$^{-/-}$ and *Casp*$^{C284A/C284A}$ bone marrow, respectively, and Vanessa Mack for technical assistance. In addition, we would also like to thank the University of Lausanne (UNIL) microscopy and proteomics core facilities for their help with data generation and analysis.

## Author Contributions

R Heilig: conceptualization, data curation, formal analysis, validation, investigation, visualization, methodology, and writing—original draft, review, and editing.

M Dilucca: resources, data curation, investigation, and methodology.

D Boucher: resources, data curation, formal analysis, investigation, methodology, and writing—original draft.

KW Chen: resources, data curation, formal analysis, investigation, methodology, and writing—original draft.

D Hancz: resources, data curation, formal analysis, investigation, methodology, and writing—original draft.

B Demarco: resources, data curation, formal analysis, investigation, methodology, and writing—original draft.

K Shkarina: resources, data curation, formal analysis, investigation, methodology, and writing—original draft.

P Broz: conceptualization, formal analysis, supervision, funding acquisition, project administration, and writing—original draft, review, and editing.

## Conflict of Interest Statement

The authors declare that they have no conflict of interest.

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
