## [Reviewer comments · Life Science Alliance]

Life Science Alliance

Caspase-1 cleaves Bid to release mitochondrial SMAC and drive secondary necrosis in absence of GSDMD

Rosalie Heilig, Marisa Dilucca, Dave Boucher, Kaiwen Chen, Dora Hancz, Benjamin Demarco, Kateryna Shkarina, and Petr Broz

DOI: <https://doi.org/10.26508/lsa.202000735>

Corresponding author(s): Petr Broz, University of Lausanne

Review Timeline:

Submission Date:	2020-04-07
Editorial Decision:	2020-04-08
Revision Received:	2020-04-14
Editorial Decision:	2020-04-15
Revision Received:	2020-04-16
Accepted:	2020-04-16

Scientific Editor: Andrea Leibfried

Transaction Report:

Please note that the manuscript was previously reviewed at another journal and the reports were taken into account in the decision-making process at Life Science Alliance. Since the original reviews are not subject to Life Science Alliance's transparent review process policy, the reports and author response cannot be published.

April 8, 2020

Re: Life Science Alliance manuscript #LSA-2020-00735-T

Prof. Petr Broz
University of Lausanne
Biochemistry
Chemin des Boveresses
Epalinges 1066
Switzerland

Dear Dr. Broz,

Thank you for transferring your manuscript entitled "Caspase-1 drives rapid secondary necrosis by inducing mitochondrial damage and SMAC release via truncation of Bid" to Life Science Alliance. The manuscript was assessed by expert reviewers at another journal before, and the editors transferred those reports to us with your permission.

The reviewers who evaluated your study elsewhere would have expected a further reaching conceptual advance and in vivo experiments in light of the recent related literature. We think that the value provided to others without such additional insight is significant, and would therefore like to publish your work here. You already provided a point-by-point response to the concerns raised by the reviewers, and I have now carefully evaluated your response. I think your response addresses the concerns in a good way, and I would thus like to ask you to upload a revised version of your manuscript, including the following changes:

Please adapt the model in figure 6 as proposed.

Please expand the Material and Methods as proposed.

Please expand the discussion to better describe the differences of your study from the one from Tsuchiya et al. (using the arguments you provide to rev#2, point 5 and 7). Please also include the additional data as proposed and discuss carefully.

Please expand the discussion on how your results could be translated into physiological settings (using the arguments you provide to rev#2, point 8).

Please mention the partial effects of the pan-caspase inhibitor on caspase-1 activity in the manuscript text.

Please change the manuscript text as proposed in response to rev#3, point 5 and minor point 10.

Please rework the scale bars as proposed.

Please reference the study by Jost et al when introducing type I and type II cells.

Because additional data will still get integrated, we will do a careful figure and manuscript check on the next manuscript version only. So I may have to ask you for additional small changes before final acceptance.

You will be guided to complete the submission of your revised manuscript and to fill in all necessary

information. Please get in touch in case you do not know or remember your login name.

Thank you for this interesting contribution to Life Science Alliance. We are looking forward to receiving your revised manuscript.

Sincerely,

B. MANUSCRIPT ORGANIZATION AND FORMATTING:

*****IMPORTANT:** It is Life Science Alliance policy that if requested, original data images must be made available. Failure to provide original images upon request will result in unavoidable delays in

publication. Please ensure that you have access to all original microscopy and blot data images before submitting your revision.***

April 15, 2020

RE: Life Science Alliance Manuscript #LSA-2020-00735-TR

Prof. Petr Broz
University of Lausanne
Biochemistry
Chemin des Boveresses
Epalinges 1066
Switzerland

Dear Dr. Broz,

Thank you for submitting your revised manuscript entitled "Caspase-1 cleaves Bid to release mitochondrial SMAC and drive secondary necrosis in absence of GSDMD". I appreciate the introduced changes and would thus be happy to publish your paper in Life Science Alliance pending final revisions necessary to meet our formatting guidelines:

- Please add a callout in the manuscript text to Fig. S1B, D; S4C,D; S6B-D
- You mention Fig. S3D on page 6, please fix
- Please label the white insets in Fig2D to make it clear to which zoomed images they correspond to; note also that one of the insets in the first image does not match the zoomed one
- Please add scale bars to Fig. S3A&B
- Please provide source data for Fig. S4D
- the MitoTracker zoomed images in some panels in Fig. S6A do not match the indicated boxed region; please fix

A. FINAL FILES:

-- High-resolution figure, supplementary figure and video files uploaded as individual files: See our detailed guidelines for preparing your production-ready images, <http://www.life-science->

alliance.org/authors

B. MANUSCRIPT ORGANIZATION AND FORMATTING:

Sincerely,

Andrea Leibfried, PhD
Executive Editor
Life Science Alliance
Meyershofstr. 1
69117 Heidelberg, Germany
t +49 6221 8891 502
e a.leibfried@life-science-alliance.org

Dear Dr. Leibfried

Please find enclosed the **revised version** our manuscript entitled "**Caspase-1 cleaves Bid to release mitochondrial SMAC and drive secondary necrosis in absence of GSDMD**" that we would like you to consider for publication in *Life Science Alliance*.

We have changed the manuscript according to your email and unloaded the new files. To address your request we have done the following:

- Please add a callout in the manuscript text to Fig. S1B, D; S4C,D; S6B-D

We have corrected the callouts

- You mention Fig. S3D on page 6, please fix

This has been fixed

- Please label the white insets in Fig2D to make it clear to which zoomed images they correspond to; note also that one of the insets in the first image does not match the zoomed one

corrected

- Please add scale bars to Fig. S3A&B

Scale bars were added

- Please provide source data for Fig. S4D

We have replaced the blots with new blots showing all data and added a .pdf with the full, uncropped blots for this panel

- the MitoTracker zoomed images in some panels in Fig. S6A do not match the indicated boxed region; please fix

corrected

April 16, 2020

RE: Life Science Alliance Manuscript #LSA-2020-00735-TRR

Prof. Petr Broz
University of Lausanne
Biochemistry
Chemin des Boveresses
Epalinges 1066
Switzerland

Dear Dr. Broz,

Thank you for submitting your Research Article entitled "Caspase-1 cleaves Bid to release mitochondrial SMAC and drive secondary necrosis in absence of GSDMD". It is a pleasure to let you know that your manuscript is now accepted for publication in Life Science Alliance. Congratulations on this interesting work.

DISTRIBUTION OF MATERIALS:

Again, congratulations on a very nice paper. I hope you found the review process to be constructive and are pleased with how the manuscript was handled editorially. We look forward to future exciting submissions from your lab.

Sincerely,

Andrea Leibfried, PhD
Executive Editor
Life Science Alliance
Meyerohofstr. 1
69117 Heidelberg, Germany
t +49 6221 8891 502
e a.leibfried@life-science-alliance.org
www.life-science-alliance.org